# COVID-19 Vaccination Rate under Different Political Incentive: A Counterfactual Trend Approach Using Nationwide Data

**DOI:** 10.3390/vaccines11071149

**Published:** 2023-06-25

**Authors:** Denis Mongin, Clement P. Buclin, Stephane Cullati, Delphine S. Courvoisier

**Affiliations:** 1Faculty of Medicine, University of Geneva, CH-1211 Geneva, Switzerland; 2Division Quality of Care, Geneva University Hospitals, CH-1211 Geneva, Switzerland; 3Population Health Laboratory (#PopHealthLab), Faculty of Science and Medicine, University of Fribourg, CH-1700 Fribourg, Switzerland

**Keywords:** COVID-19, health pass, sanitary pass, public health, vaccine acceptance

## Abstract

(1) Background: France implemented a COVID-19 certificate in July 2021 to incentivize the population to uptake COVID-19 vaccines. However, little is known about the variation in its impact across age groups and its dependence on socio-demographic, economic, logistic, or political factors. (2) Methods: Using France’s weekly first dose vaccination rate, a counterfactual trend approach allowed for the estimation of the vaccination rate across age groups at a small geographical level before and after the implementation of the health pass. The effect of the health pass was operationalized as the vaccination rate among those who would not be vaccinated without it. (3) Results: Vaccination before the health pass varied greatly among age groups and was mainly influenced by territory (lower in rural and overseas territories when compared to urban and metropolitan ones), political beliefs, and socio-economic disparities. Vaccine logistics played a minor but significant role, while the impact of COVID-19 did not affect the vaccination rate. The health pass increased the vaccination overall but with varying efficiency across groups. It convinced mainly young people politically close to the governmental vaccination strategy and living in urban metropolitan areas with low socio-economical discrepancies. The selected variables explained most of the variability of the vaccination rate before the health pass; they explained, at most, a third of the variation in the health pass effect on vaccination. (4) Conclusions: From a public health perspective, the French health pass increased the overall vaccination, but failed to promote preventive behaviours in all segments of society, particularly in vulnerable communities.

## 1. Introduction

The COVID-19 pandemic is still a large burden worldwide with direct consequences, including morbidity, mortality [1,2], and long-COVID, and indirect consequences, such as increased delay of care [3,4]. As of February 2023, the pandemic is not over, and new mutations of the virus with potentially high immunity escape are rising [5,6]. Vaccination is an important tool to mitigate COVID-19 [7], with efficient reduction in transmission, morbidity, and mortality [8], and reduction in the probability of antigenic evolution. Although most high-income countries reached a high vaccination coverage, vaccination remains a crucial subject: first, the waning of the immunity conferred by vaccines [9] forced several countries to opt for a vaccine roll-out of boosters to maintain an elevated immunity within fragile populations. Second, the rather high probability of a new epidemic event of similar intensity in the coming decades [10] means that countries may have to conduct similar vaccination programs in the future.

Vaccination is hindered by hesitancy amongst certain subgroups of the general population sharing demographic and social characteristics [11,12]. Some of these subgroups are particularly exposed to COVID-19 or its complications [13,14]. Several strategies were implemented worldwide to encourage vaccination, from simple communication and financial incentives to COVID-19 certificates. The latter was proposed early in the pandemics in the European Union, USA, and UK [15], but its effectiveness and potential side effects were the subject of intense debates [16,17]. In order to adapt and plan proper health policy strategies, it is crucial to understand the factors affecting the vaccination rates of the target population.

Numerous studies based on surveys showed that vaccination uptake depends on age, gender, socio-economic conditions such as educational achievement [14,18], psychological traits [19], fear of side effects, religious and political beliefs [20], faith in science [21,22], trust in the local or national government [21,22], and impact of the COVID-19 [14], as well as the recommendation of the person’s general practitioner and individual assumption about the immune system [23]. Although insightful, most of these studies, by design, do not examine the effect of external factors that can influence vaccination uptake, such as the supply of vaccine doses and the availability of vaccine centers or of practitioners allowed to vaccinate. Furthermore, most of them do not distinguish between the different political strategies implemented to convince the population to vaccinate. Countries’ political strategies ranged on a broad spectrum, including communication and education programs [24], more restrictive solutions, such as confinements and COVID-19 certificates, and some local governments even experimented with vaccination positive incentives, such as small cash pay-outs [25,26] or raffle participations [27]. Similarly, each country established a different prioritization list for occupations in the vaccination campaigns with different decisions, such as including or not police officers, non-healthcare workers or education staffs [28].

Other studies assessing the effect of the COVID-19 certificate [29,30,31] found an increase in vaccination rate. However, those studies did not consider the spatial variations of this effect, nor its associated factors; in particular, poverty, education, or access to healthcare. A recent study showed the univariable association of cumulative vaccination rate with political beliefs and socio-economic conditions [32], but failed to disentangle vaccination before and after the introduction of COVID-19 certificate and ignored the strong spatial correlations in their analysis.

France offers a compelling case study to examine the factors driving vaccination rates under different political incentives, as their strategy evolved over the course of the pandemic [33]. Initially, France relied on public health messages to encourage voluntary vaccination. However, this approach was later complemented by the implementation of a COVID-19 certificate (health pass) that became mandatory for non-essential leisure activities such as sports, cultural, or social events. Such incentive has an effect differing between rural or urban territories and between age and socio-economical groups. Using diffusion of innovation theory, we analyzed the weekly vaccination rates per age category at the intercommunality level (EPCI) for the 67 million residents of France. EPCI is a federation of communes, a scale aggregating communes that would be too small to be considered separately while providing geographical areas that are small enough to offer a fine discretization of the territory and of the population. Two clear separate trends were observed for the vaccination, one before the health pass and one after. Our counterfactual approach enabled us to estimate the standardized effect of the health pass (SEHP), measured as the proportion of individuals who were convinced to vaccinate because of the health pass and would not have otherwise. The objective of the present ecological study is to investigate the factors associated with vaccination rates before the health pass and with the effect of the health pass. The factors considered include socio-economic determinants, political trends, demographics characteristics, territory specificities, access to health care, and local impact of COVID-19.

## 2. Methods

The analysis of the present study consists of a spatial regression at the intercommunality scale (see geographical units) stratified per age group, studying how the vaccination before the health pass and the standardized effect of the health pass (see outcomes) are affected by being an overseas or metropolitan territory, the territory density, the COVID-19 death excess, the COVID-19 hospitalization rate, the poverty rate, the rate of population with a secondary education, the income inequalities, the minimum distance to the closest vaccination center, the number of vaccine doses delivered, the rate of abstention during the last 2022 presidential elections, and the score of the pro-health pass candidates.

### 2.1. Geographical Units

France is a diverse territory with vast cultural and geographical differences between metropolitan and rural regions, as well as overseas territories covering the many regions across the world (Latin America, Sub-Saharan Africa, and East Asia and Pacific). The chosen scale for this ecological study is the intercommunality (EPCI for Établissement public de coopération intercommunale), the smallest geographical unit with a recorded vaccination rate. The commune is the lowest level of administration in the French republic. They are governed by an elected mayor and municipal council. There are 35,965 communes in France in 2021, the sizes of which range from few inhabitants to over 2 million (Paris). The EPCI is a federation of communes, which is an administration level that allows neighbouring communes to group together to collectively handle certain local tasks and collect certain taxes. There were 1252 EPCIs in 2021, with populations ranging from 4000 people to 7 million. The upper levels of administration in France are the departments (100 departments, of which 94 are metropolitan and 6 are overseas) and the regions (19, of which 14 are metropolitan and 6 are overseas).

### 2.2. Vaccination and Its Evolution

France implemented the COVID-19 pass on 1 June 2021 and extended it on 21 July 2021 to cultural places. The whole vaccination campaign was free. France authorized the Pfizer–BioNTech COVID-19 vaccine on 22 December 2020 and the Moderna vaccine on 8 January 2021. Vaccination started on 27 December 2020, first for nursing home residents and healthcare workers (27 December 2020–18 January 2021), then it opened for the general population with decreasing age thresholds (18 January 2021–31 May 2021), and finally for the whole adult population (from 31 May 2021). France vaccination dose distribution was conducted using different strategies for different vaccines. Due to both its early availability and difficult conservation, Pfizer’s distribution was organized at the national level, prioritizing the most at-risk populations and using a dose need assessment conducted locally in nursing homes and vaccination centers. For Moderna and other vaccines, a national repartition was conducted between regions, which then transferred doses to departments based on the need of departmental health agencies (Agence Regionales de Santé) [34].

Weekly vaccination rates at the intercommunality level are provided by the national health insurance agency [35] for the different age categories: less than 20, 20–39, 40–54, 55–64, 65–74, and over 75 years old. Data are aggregated by the national health insurance from the data entered in the COVID vaccine application, which was a mandatory application since 4th of January 2021 for any medical professional performing COVID-19 vaccination. Our variable of interest is the population vaccinated with at least one dose for the age categories 20–39, 40–64, 65–74, and over 75.

In order to estimate the effect of the health pass, we calculated a counterfactual estimate of the vaccination rate without the health pass using the diffusion of innovation theory [36,37]. This theoretical framework (see Appendix A) proved to apply nicely to innovations in healthcare [29,38], and was applied in this study to the two vaccination peaks for each EPCI and each age class. The weekly first dose vaccination rate in France displays two peaks, one before and one after the implementation of the vaccine certificate. We call hereafter L1 the estimation of the first peak, and L2 the estimation of the second peak (see Figure 1). These estimations provide three characteristics of the vaccination curves: (1) the time in days until the weekly vaccination rate reaches its maximum amplitude (peak), (2) the maximum amplitude of the curve in weekly % of the population receiving the first vaccination dose, and (3) the spread of the vaccine curve over time in days (for a detailed description of the parameters, see Appendix A).

### 2.3. Outcomes

There are two outcomes, one for each of the two periods of interest. The first outcome is the vaccination rate before the COVID-19 pass: that is the cumulated percentage of the population with at least one dose of vaccination before the health pass (before the 12 July 2021). The second outcome is the standardized effect of the health pass (SEHP) that is the percentage of the population convinced to vaccinate by the health pass that would not have vaccinated otherwise. It is calculated as a ratio. In the numerator, the population convinced by the health pass (red area in Figure 1) is calculated as the cumulated percentage of the population with at least one dose after the implementation of the health pass (the area under the curve L2 in Figure 1) minus the cumulated percentage of vaccination for the counterfactual scenario without the health pass during the same period (the area under L1 in Figure 1). The denominator provides the standardization and is the percentage of the population that would not be vaccinated without the health pass, which is 100 minus the cumulated percentage of the counterfactual estimation L1 (100 minus the blue area in Figure 1).

### 2.4. Covariates

Potential determinants of the vaccination are described below.

#### 2.4.1. COVID-19 Measures

The COVID-19 measures considered are:The cumulated percentage of the population hospitalized for COVID-19 up to 12 July 2021. This measure is available at the department level [39];The death excess from the beginning of 2020 to 12 July 2021 at the EPCI level, calculated as the p-score [40]:
(1)$$p_{score}=\frac{deaths-expected\;deaths}{expected\;deaths}\times 100;$$

The details of this calculation, based on the public list of all deaths in France provided by the national institute of statistics (INSEE) [41], is available in Appendix A.

#### 2.4.2. Socio-Economic Conditions

Among the aggregated socio-economic indicators at the communal or EPCI level in France provided by INSEE, we included three variables covering social, educational, and economic aspects of the socio-economic status of the population:The rate of poverty. In France, poverty is defined as having an income below 60% that of the national median income.The rate of the population above 25 years old without secondary education.The income inequality, defined as the ratio between the higher and the lower decile of the income distribution.

A sensitivity study comparing exposure coefficients between multivariable models adjusting only for these three measures, and less parsimonious models, showed that these three measures provided a similar adjustment to models including many more socio-economic indicators.

#### 2.4.3. Access to COVID-19 Healthcare

Access to vaccination centres or health care facilities was considered using three indicators:Distance from the centre of each EPCI to the closest vaccination centre;Number of pharmacy and family physician per person in each EPCI, since both were allowed to vaccinate (although only family physicians could prescribe the vaccination);Cumulated number of vaccine doses per person at the departmental level delivered up to 12 July 2021, and cumulated number of doses per person remaining to be vaccinated during the month after 12 July.

#### 2.4.4. Territory

Geographic and demographic characteristics of the EPCIs were included by considering the population density divided into three categories: low (below 50 person/km^2^), medium (between 50 and 300 person/km^2^), and high (above 300 person/km^2^), the proportion of women, and being an overseas territory or a metropolitan territory.

#### 2.4.5. Political Leanings

Two variables were calculated from the results of the 2022 French presidential election as a measure of the political tendencies concerning vaccination and the health pass in each EPCI. First, we used the percentage of vote for a pro-pass candidate during the first round of the 2022 French election. The 11 candidates were classified according to their public declaration in 2021 as pro- or anti-health pass (see Appendix A). Second, we used the percentage of abstention during the first round of the French presidential election of 2022.

### 2.5. Statistical Analysis

All analyses were run using R4.0.0 [42]. The regressions L1 and L2 of the time evolution of the vaccination rate before and after the implementation of the health pass were performed with nonlinear least square regression.

The main analysis consisted of a spatial simultaneous autoregressive multivariate regression [43] with the spautolm function from the package spatialreg [44], using row weighted neighbours list created with the spdep package. Spatial correlations were tested using Moran’s I index.

The decomposition of variance in the outcome between the three nested geographical administrative levels (EPCIs, the departments, and the regions) was performed using the Bayesian framework with the package rjags [45] using uniform priors. All variables were normalized by centering them and dividing them by standard deviation.

Missing data were handled with multiple imputation with a chained equation with all covariates in the imputation model and the predictive mean matching algorithm, using the mice package [46]. Ten imputed datasets were analyzed separately, and the results were pooled according to Rubin’s law [47]. Data curation and handling were performed using the library data.table, and the library ggplot2 was used for the graphical representations.

All data are openly available, and data and code used in this study are provided in the following Gitlab repository https://gitlab.com/dmongin/scientific_articles/-/tree/main/vaccination_France (accessed on 20 June 2023).

## 3. Results

The innovation diffusion regressions L1 and L2 (see Figure 1) of the two vaccination curves yielded excellent fits for all age groups, with median R squared values ranging from 0.8 to 0.9 (Table 1), and it was slightly lower for the L2 regression among 75 years old or older people.

The first peak of the first dose vaccination rate (i.e., before the health pass) was closer to the health pass implementation for the younger age category, and with a smaller amplitude. The spread of the curve was especially small for the 20–39 category, with a median value of 38 days, compared to 60 for people aged between 40 and 64 years. Combining both effects, the median [inter quartile range] cumulated vaccination rate before the health pass (across the ECPI) decreased with age, from 84% [81, 86] for the population above 75 years, 82% [79, 84] for those between 65 and 75, down to 62% [58, 66] for the 40–64-year-olds and 37% [32, 42] for those between 20 and 39 years.

The second peak of weekly vaccination after the health pass implementation occurred around the same time for all age categories (around 33 days after the health pass implementation), but had a higher amplitude and lower spread for younger categories. The standardized effect of the health pass (SEHP) was 54% [48, 58] of the youngest who were not going to vaccinate, 48% [41, 54] for those between 40 and 64, 42% [34, 48] for those between 65 and 75, and 30% [17, 86] for those above 75 (Table 1).

EPCIs in metropolitan and overseas territories were especially different (Table 2). Regarding population density, 10% of the EPCIs had more than 300 inhabitants per km^2^ in metropolitan France, whereas it concerned 33% of those overseas. The median poverty rate was around 13% in metropolitan France, but rose up to 33% overseas, consistent with a lower median income and a higher inequality. These territories did not differ in term of number of family physician and pharmacy density, nor in term of vaccination centres, but overseas EPCI had a lower median number of doses delivered per capita, covering less of the unvaccinated population, and with a greater variability between the EPCI. Political results differ greatly too, as more than half of the population overseas did not vote during the first round of the 2022 presidential elections, whereas the abstention was of 22% [20, 24] in the metropolitan EPCIs.

### 3.1. Vaccination before Health Pass

The geographical repartition of the vaccination rate before the health pass is represented in Figure 2.

Focusing on cumulative vaccination rate before the health pass, the multivariable autoregressive model explained 58% of the variance between EPCIs for the 20–39 age category and up to 95% for the oldest age category (Table 3). There was an important effect of the territory: living in an overseas EPCI was strongly and negatively associated with the vaccination rate for persons above 40, the effect increasing with age: the decrease was of −14 [−20, −8] percent points (pp) for the 40–64 years category, −33 pp [−37, −29] for the 65–75 years, and −38 pp [−41, −35] for 75+. Irrespective of other covariates, the vaccination rate was higher in dense EPCIs when compared with low-density ones. This effect held for all age categories, and was stronger amongst the younger population, reaching up to almost 3 pp for the youngest categories in the densest EPCIs. The proportion of women in the EPCI had a positive and significant increase effect for the two younger age classes, with an increase of 2 pp every 10 additional percent points of women in the population, but not for the older age class.

Considering the socio-economic condition of the EPCIs, an increase in the rate of poverty was associated with a decrease in the vaccination rate, with an association decreasing with age. An increase of 10 points of the poverty rate was associated with a decrease of −5 pp [−5, −4], −4 pp [−5, −3], −2 pp [−3, −2], and −0.7 [−1, 0] of the vaccination rate for 20–39, 40–64, 65–75, and 75+ age categories, respectively. Income inequality was associated with a lower vaccination rate for ages over 40 years, with a rather similar effect across the age concerned: an increase of 1 in the ratio between the upper and the lower decile of salaries decreased the vaccination rate by 1.7 pp.

The political belief of the population was a strong predictor of the vaccination rate, with a similar effect for all age categories. An increase of 10 points of the abstention percentage in the 2022 presidential election decreased the vaccination rate by 2 pp. An increase of 10 percent points for candidates that are pro-vaccination increased by 5 pp for the vaccination rate below 65 years old and by 3 pp for above 65.

None of the COVID-19 consequences (death excess or hospitalization) were associated with the vaccination rate, in contrast to the variables concerning health access. The distance to a vaccination centre (by each 10 km from closest centre) decreased the vaccination rate by −0.4 pp [−0.6, −0.3] for people 20–39 years old and −0.5 pp [−0.8, −0.3] for 40–64 years old. The density of physicians and pharmacies did not influence the vaccination rate, but the number of doses delivered did: an increase of one dose per person was associated with an increase of 3 pp [1, 5], 2 pp [0, 5], and 1 pp [0, 2] for the 40–64, 65–75 and 75+ age categories, respectively.

The main contributions among the significant effects were related to the territory, namely living in an overseas or a dense territory, the rate of poverty for the age categories below 65 years, and political belief (Appendix A, normalized coefficients, Appendix A).

### 3.2. Vaccination after Health Pass

The geographical repartition of the standardized effect of the health pass is represented in Figure 3. Same as the vaccination rate, a strong spatial correlation was observed for the two youngest age categories, with a significant Moran index of 0.7 and 0.6, respectively (ps < 0.001), while the 65–75 years and 75+ age categories did not display a clear spatial correlation and had a Moran index of only 0.1 (ps < 0.001).

The multivariable autoregressive model explained 41% of the variance of the SEHP for the youngest age category, but only 15% for the 65–75 age category (Table 4). 

Living in an overseas territory strongly decreased the SEHP, especially for the 75+ age category (−39 pp [−50, −28]). Irrespective of other covariates, SEHP increased significantly in mildly or highly dense territories for older people (+5 pp for the 65–75 age category, and > +10 pp for those over 75 years). The proportion of women in the EPCI had a positive effect on SEHP, with an increasing effect as age increased, from 1.2 pp [0.7, 1.7] for people 40–64 years old to 3.8 pp [2.7, 4.8] for those 75+.

The proportion of people hospitalized for COVID-19 was associated with a lower SEHP for the 65–75 and 75+ categories, with −5 pp [−8.6, −1.4] and −7 pp [−10, −2], respectively, for each increase of 1 pp of people hospitalized for COVID-19.

Socio-economic factors were mostly associated with SEHP for the younger age class. For instance, every 10 pp of poverty decreased SEHP by −2 pp [−3, −1] and −3 pp [−5, −1] for those 20–39 and 40–64 years old. Similarly, for people among the 65–75 years category, an increase of 1 in the ratio between the upper and lower decile of income (income inequality) was associated with a decrease of −5 pp [−7, −2] of SEHP. Finally, SEHP also decreased with the proportion of secondary education diplomas, but only in those above 65 years old.

Access to health care played a marginal role, with small effects of the distance to a vaccination center and the number of available doses. However, compared to the vaccination rate before the health pass, the density of GP and pharmacies showed a stronger association with a lower SEHP for 40–64 (−3.9 pp in SEHP).

Concerning the political trends, abstention was associated with a lower SEHP for those 20–39 years old, with a reduction of −1 pp of the standardized effect of the health pass every 10 additional points of abstention, but had the opposite effect for the older age categories: An increase in abstention of 10 points was associated with an SEHP increase of 3 pp [0.4, 5], 4 pp [2, 7], and 5pp [2, 7] for the 40–64, 65–75, and 75+ age categories, respectively. The percentage of votes for a pro-pass candidate was associated with a higher SEHP globally, but only reaching significance for the 20–39 and the 65–74 age categories.

The main contributions among the significant effects were not the same for the different age categories (Appendix A, normalized coefficients, Appendix A). For people above 65 years, the variables having the most impact were related to the territory, i.e., the density of the EPCI or metropolitan versus overseas, while for the youngest, the political and societal variables were the most important.

## 4. Discussion

In this ecological and nationwide study of the COVID-19 vaccination rate at a small geographical level (ECPI), we used a counterfactual trend approach to estimate the impact of the health pass on vaccination rates in France. We also examined the associations of age, geographical data, socioeconomics, political orientations, COVID-19 vaccination facilities access, and COVID-19 impact with vaccination before and after the introduction of the health pass in France. As observed elsewhere [29,30,31], we found that the health pass increased the vaccination, but with differing impact across age groups. It managed to convince 54% of the youngest that would not be vaccinated without it, but only 30% of the oldest. On the other hand, before the implementation of the health pass, the vaccination rate was of 37% for the youngest age category and 84% for the oldest one. Although selected variables explained most of the variability of the vaccination rate before the health pass, they explained at most a third of the variation in the health pass effect on vaccination.

The fact that the variables explaining most of the vaccination rate before the health pass did not explain more than a third of the health pass effect variability across the country indicates that there are other determinants when vaccination occurs under a coercive environment that are not captured by our analysis. These could be related to use of social media [18,48], religion [20,49], or other factors that did not affect vaccination without the health pass. Further research into factors explaining variability of health pass effectiveness is needed to shed light on these factors and allow for an optimized management of such a health policy.

Living in overseas territory had a major effect on vaccination both before and after the health pass. Overseas territories were much less vaccinated and less incited to vaccinate by the health pass, even when adjusting for poverty rate, income inequality, and access to vaccine doses. This decrease in both the vaccination rate before health pass and the health pass effect was much stronger for older age categories. Indeed, for the more vulnerable age category (75+), it reached −38 percent points of the vaccination rate before the health pass, and −39 percent points for the percentage of persons convinced by the health pass who would not be vaccinated otherwise. Overseas territories of France are territories where people can experience a feeling of exclusion from the social state, as they suffer a high and long-lasting level of unemployment [50], a high rate of poverty, important problems of access to drinking water [51], or recurring environmental contaminations [52]. This could explain the important difference observed for these territories, and potentially its association with age, as younger people may be less inclined to this feeling because of recent active efforts from the French government to palliate the above-cited problems. Overseas territories may also have different ethnical backgrounds compared to metropolitan France, but no data were available to control for this factor.

The density of the territory was also associated with both the vaccination rate before the health pass and the health pass effect, with an effect stronger for young age categories for the first and stronger for older age categories for the latter. This result is in line with previous studies showing a difference in vaccination rates in urban and rural areas [53]. There are more crowded places in dense territories, which potentially favoured vaccination before the health pass because of a higher need to protect themselves and others from contamination. Similarly, the increased health pass effect could be a consequence of the higher number of social places and events requiring a valid health pass. Another explanation of the association between population density and vaccination rate may be the access to technology. Indeed, vaccination appointment was made in France through a website, which could be a limitation for people having difficulties with technologies or having poor internet connection, which is the case for around 30% of the French population, mainly aged and located in the countryside [54]. Other European countries, such as Portugal and Spain, did not use a web-based application for vaccine appointments, but instead organized at the state level the vaccination roll-out plan using the exhaustive list of the national insurance to make appointments and reached a high level of vaccine coverage of the population in a few months.

Regional poverty and income inequalities were both associated with lower vaccination before the health pass and a reduced effect of the health pass, as observed in France [55], in Switzerland [56,57], and elsewhere [14,21]. The inverse care law [58], stating that deprived territories needing the most care are likely to receive the least, may partly explain this association through access to the health system in general [21,59,60]. Interestingly, when adjusting for poverty and income inequalities, education was no longer associated with vaccination.

The proportion of women was associated with vaccination rates, which may seem at first to be in contradiction with various studies observing a higher vaccine hesitancy for women [13,19,61,62,63]; but whereas these studies examined intention to vaccinate, our study observed the actual vaccination uptake. A Danish nationwide study using actual vaccination from healthcare registries showed higher uptake in vaccination among women [64]. Another study in the UK and Ireland reported that the gender difference disappeared when considering vaccine refusal instead of vaccine hesitancy [19]. The positive association between percentage of women and vaccination could be explained by the higher prevalence of women in family care-related tasks (children, caregiving of old parents, etc.) and their higher propensity to perceive threats [65] or respect injunctions [66], counterbalancing their higher hesitancy.

Interestingly, the regional mortality (excess deaths) and regional morbidity (hospitalizations) of COVID-19 had little to no effect on vaccination. The only effect observed concern the old age categories, with a slight reduction in vaccination before the health pass and a small reduction in the pass effect on convincing people to vaccinate. Given the age category concerned, these small associations could be inverse causal relations, where less-vaccinated EPCI undergo higher mortality.

The logistics surrounding the vaccination had its importance. First, having a vaccination centre nearby increased the vaccination coverage before the health pass for young adults and adults, proving that providing facilities allowing vaccination without appointment was effective. Second, a higher number of doses delivered was also associated with higher vaccination rate for old age categories, which potentially reduced saturated appointment services and discouragement, especially for the persons the most refractive to technology.

Vaccination before the health pass was lower in territories where the abstention was higher and in territories where candidates against the health pass had a high score. This was especially the case for the working age population, and confirms the results of various survey studies [19,20,67]. The effect of the health pass was higher for people above 40 in territories with a higher rate of abstention, which may indicate that people feeling excluded from politics were less inclined to vaccinate but more responsive to coercive measures. The effect of the health pass was higher in regions electing politicians supporting the health pass but to a lesser extent than before the health pass, which may be because people agreeing with this measure were already vaccinated. The strong influence of the political variables in our study, as well as in a meta-analysis on vaccination uptake in the United States [12], indicates that the trust and the implication of the population in the political system, together with the global messages given by the political class, are of prime importance for the success of such a campaign.

The strengths of the present study lie in the use of France’s exhaustive vaccination data, including both mainland and overseas territories, hence representing a large variability of territories and populations whilst keeping the same vaccination and health pass strategies. The statistical analysis accounting for the strong spatial correlation, separating with a proper counterfactual modelling the two different vaccination periods, and using a multivariable analysis on a large set of preselected variables, also strengthen the findings. Nevertheless, there is certainly residual confounding due to unmeasured variables. Indeed, information on religion/belief [20], risk perceptions of COVID-19 infection, or perceptions of vaccine safety [68], were not included, although they were proven to influence vaccination. A second limitation is the risk of ecological fallacy inherent to approaches based on spatially aggregated data, where an association could be caused by individual characteristics, not captured at the local level. A last limitation is the measurement error associated with the data. Indeed, data are provided by departmental, regional, and national state agencies, which are not exempt of errors. Although this effect could be differential between poor or rich regions, we expect the global amount of error to be low in a developed country with a long administrative tradition, such as France, thus having little impact on our results.

## 5. Conclusions

Overall, vaccination without mandate was mainly affected in France by the political debate, logistics, income inequalities, and the variation in population density across the country. The health pass, thought to force the vaccine-hesitant population to vaccinate, was effective in convincing an important part of the young population to vaccinate, but failed in convincing the majority of the vulnerable unvaccinated and in correcting the socio-economic disparities of the initial vaccination program. Thus, from a public health perspective, the French health pass failed to achieve its goal of controlling communicable diseases by promoting preventive behaviours in all segments of society and, in particular, in vulnerable communities.

## Figures and Tables

**Figure 1 vaccines-11-01149-f001:**
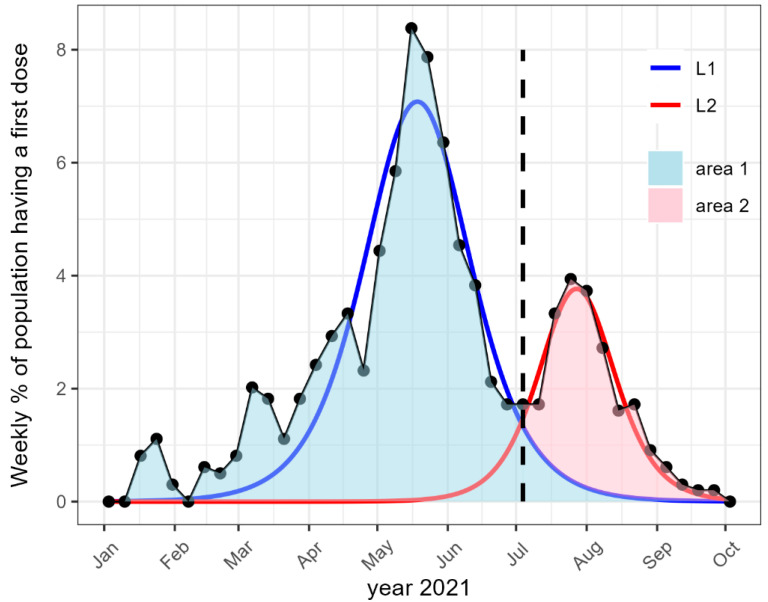
Illustration of the counterfactual approach used to analyze the weekly vaccination rate (dots) of a given EPCI and age category. The blue line and the red lines represent the fits L1 and L2 of the two peaks obtained with the innovation diffusion model. The blue area represents the counterfactual estimation of the cumulative percentage of the population without COVID-19 certificate. The red area is the estimated additional vaccine uptake due to the COVID-19 certificate.

**Figure 2 vaccines-11-01149-f002:**
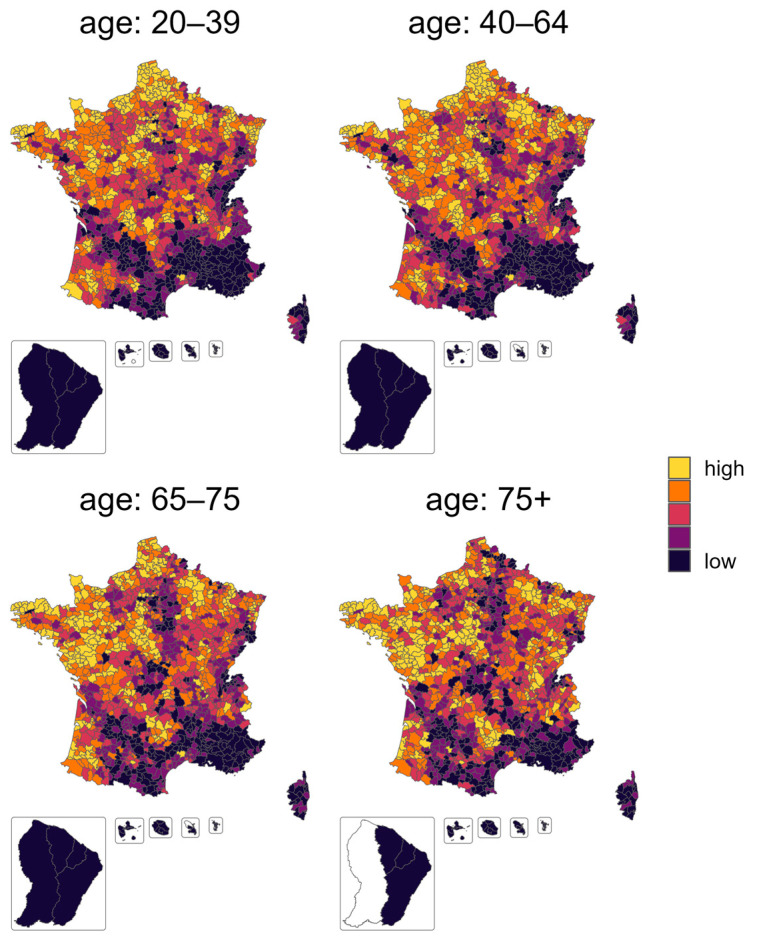
Geographical repartition of the vaccination rate before the health pass. The vaccination rate is represented per quintile category in order to observe the spatial clustering of the variable.

**Figure 3 vaccines-11-01149-f003:**
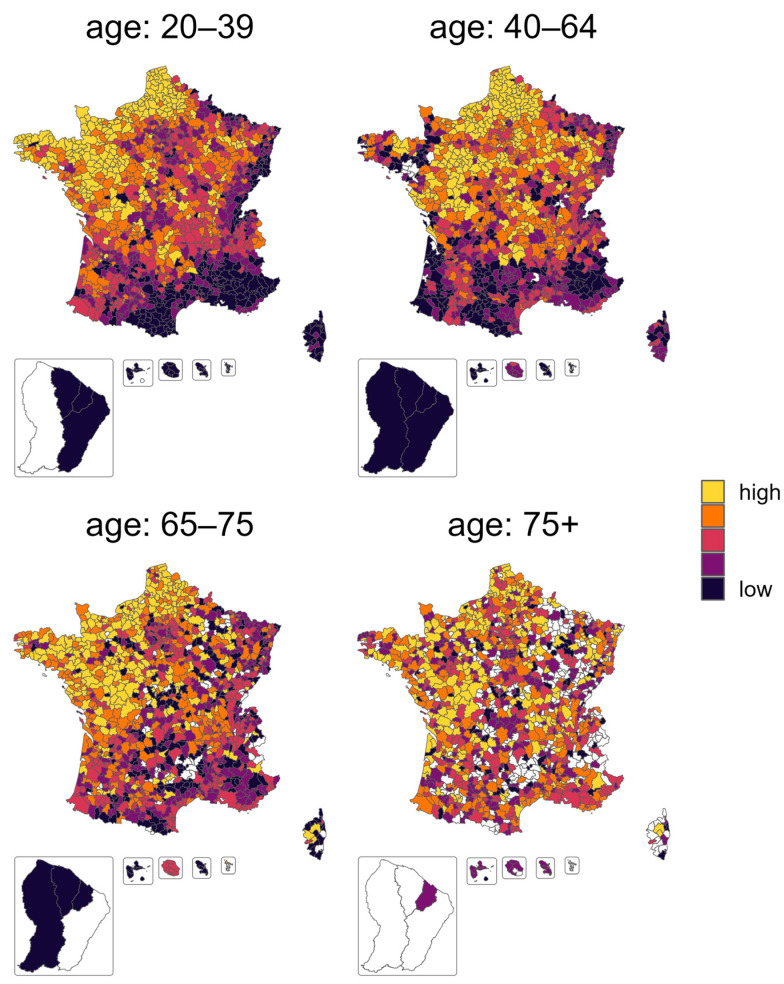
Geographical repartition of the SEHP, that is the percentage of hesitant people convinced by the health pass. This variable is represented per quintile category in order to observe the spatial clustering of the variable. Missing variables are colored in white.

**Table 1 vaccines-11-01149-t001:** Median [IQR] values for the two outcomes considered, for the parameters of peak yielded by the innovation diffusion regression L1 and L2 of the weekly vaccination rate, by age groups. The time $t_{max}$ is the time between the implementation of the health pass and the peak maximum, the amplitude of the peaks is the maximum weekly vaccination rate obtained, and the width γ corresponds in days to the full width at half maximum of the vaccination rate peak. R^2^ gives the R square of the regression. SEHP: standardized effect of the health pass, which is the percentage of the population convinced to vaccinate by the health pass that would not be vaccinated otherwise.

Age Class	20–39	40–64	65–75	75+
Outcomes				
Vaccination rate before health pass	37.3 [32.3, 41.9]	62.0 [57.7, 66.0]	81.6 [78.7, 84.1]	84.0 [81.4, 86.1]
SEHP	53.6 [48.6, 58.1]	48.6 [41.5, 54.7]	41.6 [34.2, 48.0]	30.1 [16.6, 37.6]
**Fit L1**				
Time $t_{max}$ in day from the health pass	−21.3 [−25.5, −17.8]	−37.9 [−41.9, −34.2]	−70.9 [−74.9, −67.8]	−92.8 [−97.5, −88.3]
Amplitude Vmax in weekly % of the population receiving first vaccination dose	6.1 [5.1, 7.1]	6.4 [5.7, 7.2]	9.3 [8.4, 10.4]	8.7 [7.3, 10.3]
Full width at half maximum γ	37.9 [33.9, 43.1]	60.2 [51.4, 70.6]	53.4 [46.5, 60.4]	44.1 [33.7, 56.1]
R^2^ of the regression	0.93 [0.90, 0.96]	0.84 [0.76, 0.89]	0.84 [0.77, 0.90]	0.82 [0.74, 0.88]
**Fit L2**				
Time $t_{max}$	32.7 [30.7, 35.0]	32.5 [30.4, 34.7]	31.9 [29.3, 34.9]	34.9 [30.0, 39.1]
Amplitude	6.4 [5.8, 7.1]	3.6 [3.3, 4.0]	1.3 [1.1, 1.6]	0.8 [0.6, 1.0]
Width γ	35.8 [31.7, 40.6]	40.1 [35.2, 45.0]	43.3 [31.2, 51.1]	47.1 [26.4, 64.0]
R^2^	0.92 [0.88, 0.94]	0.92 [0.88, 0.95]	0.84 [0.75, 0.90]	0.71 [0.50, 0.84]

**Table 2 vaccines-11-01149-t002:** Median [IQR] values or number (percentage) of the EPCI variables considered in the multivariable analysis, stratified between metropolitan and overseas EPCIs.

	Overall	Metropolitan	Overseas	Missing
Territory				
Number of EPCI	1252	1229	23	
Density				0.4
Low (below 50/km^2^)	508 (40.7%)	504 (41.0%)	4 (22.2%)	
Medium (between 50/km^2^ and 300/km^2^)	611 (49.0%)	603 (49.1%)	8 (44.4%)	
High (above 300/km^2^)	128 (10.3%)	122 (9.9%)	6 (33.3%)	
COVID-19				
Death excess (p score)	8.70 [1.46, 17.26]	8.62 [1.31, 17.24]	10.75 [6.73, 22.62]	0.0
hospitalization (% of the population)	0.67 [0.45, 0.85]	0.67 [0.45, 0.85]	0.52 [0.28, 0.52]	0.5
Socio-economics				
population below the poverty threshold (%)	12.90 [9.70, 16.00]	12.90 [9.70, 15.90]	33.55 [30.58, 38.78]	1.3
population without a secondary education (%)	17.50 [16.60, 18.60]	17.60 [16.70, 18.60]	15.90 [15.15, 17.58]	0.4
Income inequality	2.80 [2.70, 3.10]	2.80 [2.70, 3.10]	4.15 [3.98, 4.32]	1.3
Median living (k euros)	21.10 [20.20, 22.30]	21.10 [20.28, 22.30]	16.85 [15.45, 17.65]	1.3
COVID-19 health access				
Epci with a vaccination center (%)	930 (74.3%)	911 (74.1%)	19 (82.6%)	0.0
Distance to the closest vaccination center (km)	5.85 [3.17, 10.51]	5.75 [3.15, 10.40]	10.60 [6.90, 19.17]	0.0
Number of family physicians and pharmacies per 10,000 inhabitants	1.1 [0.9, 1.3]	1.1 [0.9, 1.3]	1.2 [0.9, 1.3]	0.6
Doses delivered before the health pass (% of the population)	90.0 [85.0, 94.6]	90.0 [85.0, 94.6]	82.9 [75.7, 134.2]	0.1
Doses delivered to vaccination centers after health pass implementation (% of unvaccinated people)	124.5 [109.4, 140.5]	125.3 [109.9, 141.4]	60.1 [12.3, 76.1]	0.1
Politics				
Percentage of abstention during first round of 2022 presidential elections	22.42 [20.34, 24.62]	22.35 [20.31, 24.42]	56.54 [52.44, 60.47]	0.0
Percentage of voting for a pro-pass candidate	31.25 [28.14, 35.02]	31.37 [28.36, 35.08]	21.16 [17.98, 22.33]	0.0

**Table 3 vaccines-11-01149-t003:** R square, intercept, and coefficients [confidence interval] of the multivariate regression predicting the cumulated percentage of the population vaccinated before the implementation of the health pass for each age category; Coefficients with confidence interval not encompassing zero are highlighted in bold.

Term	20–39	40–64	65–75	75+
R^2^	0.58	0.81	0.94	0.95
**Territory**				
Oversea (vs metropolitan)	4.0 [−5.1,13.1]	**−14.2 [−20.2,−8.1]**	**−32.7 [−36.8,−28.6]**	**−38.0 [−41.1,−35.0]**
Medium density (ref low)	**1.3 [0.8,1.7]**	**1.2 [0.8,1.7]**	**0.8 [0.4,1.2]**	**0.5 [0.07,0.9]**
High density (ref low)	**2.9 [2.1,3.7]**	**2.4 [1.6,3.1]**	**1.7 [1.1,2.4]**	**1.4 [0.8,2.0]**
Proportion women	**0.2 [0.02,0.4]**	**0.2 [−0.002,0.4]**	0.06 [−0.1,0.2]	−0.08 [−0.2,0.09]
**COVID-19**				
Death excess	−0.004 [−0.01, 0.004]	−0.005 [−0.01, 0.003]	**−0.008 [−0.01, −0.001]**	**−0.01 [−0.02, −0.004]**
Hospitalization	0.9 [−1.1, 2.8]	−0.6 [−2.7, 1.5]	−0.5 [−2.4, 1.5]	−0.3 [−1.2, 0.6]
**Socio-economic**				
Poverty	**−0.5 [−0.53, −0.4]**	**−0.4 [−0.5, −0.3]**	**−0.2 [−0.3, −0.16]**	**−0.07 [−0.1, −0.02]**
Population without secondary education	−0.1 [−0.2, 0.03]	**−0.1 [−0.3, −0.01]**	−0.03 [−0.1, 0.08]	−0.01 [−0.1, 0.09]
Inequalities	−0.5 [−1.2, 0.3]	**−1.7 [−2.5, −0.9]**	**−1.7 [−2.4, −1.0]**	**−1.4 [−2.0, −0.8]**
**COVID-19 health access**				
Min distance to vaccination center	**−0.04 [−0.06, −0.03]**	**−0.05 [−0.08, −0.03]**	−0.01 [−0.03, 0.007]	−0.006 [−0.02, 0.01]
Pharmacy and physician density	−0.3 [−0.7, 0.1]	0.2 [−0.1, 0.6]	0.2 [−0.1, 0.5]	0.1 [−0.2, 0.5]
Doses available	−0.005 [−0.02, 0.01]	**0.03 [0.01, 0.05]**	**0.02 [0.007, 0.04]**	**0.01 [0.002, 0.02]**
**Politics**				
Abstention	**−0.2 [−0.3, −0.1]**	−0.2 [−0.25, −0.1]	**−0.1 [−0.2, −0.06]**	**−0.2 [−0.24, −0.1]**
Result of pro-pass candidate	**0.5 [0.4, 0.55]**	**0.5 [0.46, 0.6]**	**0.3 [0.27, 0.4]**	**0.3 [0.31, 0.4]**

**Table 4 vaccines-11-01149-t004:** R square, intercept, and coefficients [confidence interval] of the multivariable regression predicting the standardized effect of the health pass (the percentage of the population convinced by the health pass that would not be vaccinated otherwise) for each age category; Coefficients with confidence interval not encompassing zero are highlighted in bold.

Term	20–39	40–64	65–75	75+
R^2^	0.32	0.18	0.14	0.23
**Territory**				
Oversea (vs metropolitan)	−2.1 [−7.6, 3.3]	**−28.6 [−38.5, −18.8]**	**−21.2 [−30.4, −12.0]**	**−39.1 [−50.3, −27.9]**
Medium density (ref low)	0.4 [−0.2, 0.9]	0.5 [−0.7, 1.7]	**5.0 [3.2, 6.7]**	**10.6 [8.4, 12.7]**
High density (ref low)	0.5 [−0.5, 1.4]	0.8 [−1.2, 2.8]	**5.3 [2.3, 8.2]**	**9.9 [6.4, 13.4]**
Proportion women	0.2 [−0.07, 0.4]	**1.2 [0.7, 1.7]**	**2.4 [1.5, 3.3]**	**3.8 [2.7, 4.8]**
**COVID-19**				
Death excess	−0.01 [−0.02, 0.002]	0.002 [−0.02, 0.03]	**0.08 [0.02, 0.1]**	−0.03 [−0.09, 0.04]
Hospitalization	−1.7 [−3.4, 0.002]	1.0 [−2.7, 4.6]	**−5.0 [−8.6, −1.4]**	**−6.6 [−10.3, −2.8]**
**Socio-economic**				
Poverty	**−0.2 [−0.3, −0.1]**	**−0.3 [−0.5, −0.1]**	0.1 [−0.2, 0.4]	0.2 [−0.1, 0.5]
Pop without secondary education	−0.1 [−0.3, 0.06]	−0.2 [−0.6, 0.1]	**−0.9 [−1.4, −0.4]**	**−1.6 [−2.2, −1.0]**
Inequalities	**−3.2 [−4.1, −2.2]**	**−3.2 [−5.1, −1.2]**	**−5.0 [−7.5, −2.5]**	−1.7 [−4.4, 1.1]
**COVID-19 health access**				
Min distance to vaccination center	−0.008 [−0.04, 0.02]	−0.03 [−0.08, 0.03]	**−0.08 [−0.2, −0.003]**	−0.07 [−0.2, 0.02]
Pharmacy and physician density	−0.5 [−1.1, 0.1]	**−3.9 [−5.1, −2.6]**	−1.6 [−3.8, 0.6]	−2.6 [−5.2, 0.06]
Doses available	**0.03 [0.02, 0.05]**	0.002 [−0.03, 0.04]	**0.05 [0.01, 0.09]**	0.005 [−0.04, 0.05]
**Politics**				
Abstention	**−0.1 [−0.2, −0.02]**	**0.3 [0.04, 0.5]**	**0.4 [0.2, 0.7]**	**0.5 [0.2, 0.7]**
Result of pro-pass candidate	**0.3 [0.28, 0.4]**	0.1 [−0.01, 0.3]	**0.3 [0.1, 0.5]**	0.2 [−0.06, 0.4]

## Data Availability

All the data are openly available, and all the data and code used in this study is provided in the following Gitlab repository https://gitlab.com/dmongin/scientific_articles/-/tree/main/vaccination_France (accessed on 20 June 2023).

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
