# Peer review of "COVID-19 Vaccination Rate under Different Political Incentive: A Counterfactual Trend Approach Using Nationwide Data"

_vaccines, 2023, doi:10.3390/vaccines11071149_

Round 1

Reviewer 1 Report

Thank you for sharing your well structured and well written article on the COVID-19 vaccination rate in France following the implementation of the so called health pass. Here some very minor comments that could help to improve the article.

L78-80: You stated that the COVID-19 certificate or the so called health pass became mandatory also for non-essential leisure activities such as sports, cultural or social events. Please state more clearly in your manuscript how this fact has impacted/biased your second outcome as stated in L148-150.

L82: I am wondering which measures you took to capture once/multiple times the 67 million residents of France for data assessment, basically your sampling strategy. Please incorporate in your manuscript. 

Author Response

Please, see our answer in the file attached

Reviewer 2 Report

Estimated Authors,

I've read with great interest the present ecological study on the COVID-19 vaccination rates in France. In this study, Authors do provide an analysis of the factors associated with vaccination rates / vaccine hesitancy in France during the COVID-19 vaccination campaign, specifically targeting the use and the role of COVID-19 pass (or Green Pass in other European Countries).

Authors have addressed this topic through a very elegant but also quite complex approach that is accurately described in the methods section. In this regard, I've faced several difficulties in properly catching your text because of the following issues (that I will therefore recommend to fix):

1) the definition of EPCI in France should be detailed in the introduction section, as early as possible, explaining why your study has specifically targeted EPCI rather than other local administrative units - not only stressing the actual availability of data, but also the vantages associated with this choice.

2)  the description of the actual limits of this study should be expanded in order to encompass the reliability of the included data;

3) Please include a "conclusion" section in order to properly address specifical endpoints of this study;

4) I've not understood why did you report in Fig. 1 "CA du Pays de Laon". I was quite confused and therefore I'm recommending a fixation of this point across the main text

the main paper is largely designed with accuracy and well written:

I've no further requests

Author Response

please, see our comments in the file attached

Round 2

Reviewer 1 Report

Thank you for sharing the revised manuscript. All comments were addressed sufficiently.